# Nationwide Burden of Metallo-β-Lactamase Genes in Brazilian Clinical *Klebsiella pneumoniae* Isolates: A Systematic Review and Meta-Analysis

**DOI:** 10.3390/antibiotics14090951

**Published:** 2025-09-19

**Authors:** Carolynne Silva dos Santos, Marcos Jessé Abrahão Silva, Pabllo Antonny Silva dos Santos, Emilly Victória Correia de Miranda, Ana Beatriz Tavares Duarte, Caio Augusto Martins Aires, Luana Nepomuceno Gondim Costa Lima, Danielle Murici Brasiliense, Cintya de Oliveira Souza, Karla Valéria Batista Lima, Yan Corrêa Rodrigues

**Affiliations:** 1Program in Parasitic Biology in the Amazon Region (PPGBPA), State University of Pará (UEPA), Tv. Perebebuí, 2623-Marco, Belém 66087-662, PA, Brazil; carollyne83@hotmail.com (C.S.d.S.); antonnypabllo@gmail.com (P.A.S.d.S.); luanalima@iec.gov.br (L.N.G.C.L.); daniellemurici@iec.gov.br (D.M.B.); karlalima@iec.gov.br (K.V.B.L.); 2Bacteriology and Mycology Section, Evandro Chagas Institute (SEBAC/IEC), Ministry of Health, Ananindeua 67030-000, PA, Brazil; emillycm.bio@outlook.com (E.V.C.d.M.); beatriztd2003@gmail.com (A.B.T.D.); cintyaoliveira@iec.gov.br (C.d.O.S.); 3Department of Health Sciences (DCS), Federal Rural University of the Semi-Arid Region (UFERSA), Av. Francisco Mota, 572-Bairro Costa e Silva, Mossoró 59625-900, RN, Brazil; caio.aires@ufersa.edu.br; 4Microbiology Laboratory, Serra Talhada Academic Unit (UAST), Federal Rural University of Pernambuco (UFRPE), Av. Gregório Ferraz Nogueira, Serra Talhada 56909-535, PE, Brazil

**Keywords:** metallo-beta-lactamases, meta-analysis, Beta-lactamases, Brazil, *Klebsiella pneumoniae*, carbapenemases

## Abstract

**Background**: Class B carbapenemases confer high-level resistance to carbapenems in *Klebsiella pneumoniae*. In Brazil, data on the national burden and geographic distribution of these genes among clinical *K. pneumoniae* isolates are sporadic. We performed a systematic review and meta-analysis to estimate the prevalence of MβL genes in Brazilian clinical *K. pneumoniae*. **Methods**: We searched SciELO, PubMed, ScienceDirect and LILACS for original studies published between 2006 and 2024 reporting molecular detection of MβL in clinical *K. pneumoniae* isolates from Brazil. Articles were independently screened, along with the extracted data and appraised study quality using Joanna Briggs Institute checklists. A random-effects meta-analysis estimated the pooled prevalence of MβL producers and assessed heterogeneity and publication bias. **Results**: Fifteen studies including 3.533 clinical *K. pneumoniae* isolates met inclusion criteria. Overall, 402 isolates (11.4%) harbored MβL genes, yielding a pooled prevalence of 44.6%. Subgroup analysis demonstrated highest prevalence in the Southeast. *bla*_NDM_ was the dominant variant (present in 14/15 studies), with *bla*_VIM_ and *bla*_IMP_ rarely detected. Meta-regression revealed an inverse association between sample size and reported prevalence, and no significant publication bias was observed. **Conclusions**: MβLs, particularly NDM, are widespread in Brazilian clinical *K. pneumoniae* but show marked regional heterogeneity driven by differences in study design, laboratory capacity, and outbreak dynamics. Urgent expansion of standardized and multicenter molecular surveillance, including allele-specific detection, and strengthened laboratory infrastructure are needed and may inform targeted infection-control and antimicrobial-stewardship interventions.

## 1. Introduction

*Klebsiella pneumoniae* is a Gram-negative bacterium within the *Enterobacteriaceae* family, recognized as one of the most significant pathogens associated with both community-acquired and healthcare-associated infections (HAIs). This pathogen demonstrates extensive genetic and ecological diversity, thriving in natural environments such as soil, water, plants, and livestock, in addition to human and animal reservoirs [1].

The species is a leading cause of severe illnesses, including pneumonia, bloodstream infections, urinary tract infections, and surgical site infections, especially among immunocompromised individuals. The global health concern surrounding *K. pneumoniae* stems from its remarkable ability to develop multidrug and extensive resistance (MDR/XDR) and its inclusion in the critical-priority ESKAPE pathogens list, a group notorious for evading antimicrobial therapies and causing significant challenges in infection control and treatment [2,3].

The increasing frequency of outbreaks caused by carbapenem-resistant *K. pneumoniae* (CR-KP) highlights the urgent need for effective control and surveillance measures. Carbapenem resistance primarily mediated by β-lactamase enzymes, such as carbapenemases, severely limits treatment options and increases mortality, length of hospital stays, and healthcare costs [4,5]. Historically, β-lactamases have been classified based on the proposals of Ambler and Bush, Jacoby, and Medeiros [6,7]. Ambler divided β-lactamases into molecular classes (A, B, C, and D), according to the molecular structure of the enzyme (protein), thus being known as Ambler’s molecular class, probably the most widely used and well-known classification. The enzymes belonging to classes A, C, and D are serine-β-lactamases, which have a serine amino acid in the active center of the enzyme. Class B enzymes are metallo-β-lactamases (MβLs), which are dependent on a metal (usually zinc) as a cofactor for enzyme activity.

MβLs represent a critical resistance mechanism in *K. pneumoniae*. These enzymes hydrolyze virtually all β-lactam antibiotics, including carbapenems, which are typically reserved as last-resort therapies against Gram-negative infections [8]. Key MβL variants include New Delhi metallo-β-lactamase (NDM), imipenemase (IMP), and Verona integron-encoded metallo-β-lactamase (VIM). Alarmingly, NDM was first identified in *K. pneumoniae* in 2008 and has since achieved global dissemination, including in Brazil. Since its initial detection there in 2013, NDM-producing *K. pneumoniae* strains have emerged as an escalating threat owing to their rapid spread and high-level resistance [9,10,11].

Although efforts to detect and report MβL-producing strains persist, available and comprehensive data remain scattered across sporadic publications and often fail to capture Brazil’s diverse geographic and socioeconomic landscapes. This systematic review and meta-analysis aim to minimize this gap by compiling national prevalence data on class B carbapenemase-producing *K. pneumoniae*. By providing a thorough, evidence-based evaluation, the study provides a detailed epidemiological baseline to guide future surveillance efforts and targeted control measures.

## 2. Material and Methods

### 2.1. Study Design

This is a systematic review and meta-analysis that aims to investigate data on antimicrobial resistance (AMR) related to the class B carbapenemases’ (MβLs) genetic mechanisms in *K. pneumoniae* isolated in the Brazilian territory. This study followed the recommendations of the Preferred Report Items of a Systematic Review and Meta-Analyses (PRISMA) and was registered with the International Prospective Register of Systematic Reviews (PROSPERO) under registration number CRD420251110931 [12]. A proportion meta-analysis was conducted in this study to investigate the prevalence of MβLs.

### 2.2. Search Strategy and Eligibility Criteria

The POT strategy was used to construct the guiding question for this systematic review, which consisted of the following: “What is the prevalence of class B carbapenemases in Brazilian clinical related samples of *K. pneumoniae*?”. The anagram for its formation comprised “P” for problem (Carbapenemases-CR-KP), “O” for outcome (frequency of class B carbapenemases-KP in Brazil), and “T” for type of study (original studies) [13,14]. The keywords used to search the databases were as follows: Beta-lactamases; Brazil; *Klebsiella pneumoniae*; Metallo-beta-lactamases; Bacteria; Carbapenemases (AND).

Literature searches were performed in SciELO, PubMed, ScienceDirect, and LILACS—databases that together encompass most of the Brazilian scientific research. Primary studies published in Portuguese, English or Spanish from January 2006 through December 2024 were eligible if they employed analytical, case-control, cohort or cross-sectional designs. Short communications, letters to the editor, editorials, and any articles not freely available in full text were excluded.

### 2.3. Data Extraction and Assessment of Methodological Quality

The data was collected from all the databases mentioned above and distributed in an organized form in a computer spreadsheet application (Microsoft Office Excel 365 v.16.89.1). The database search, collection, investigation, tabulation, and data extraction were carried out independently by two authors (CSS and MJAS). Any inconsistencies between the analyses were resolved with the help of a third researcher (YCR). The data extracted from the articles were authors, year of publication, title, source(s), methodology, sample type, Brazilian city/region, collection method, setting profile, gene identification method, period in which the *K. Pneumoniae* isolates were collected, and the appropriate frequencies of class B-KP carbapenemases. Also, studies included in the present meta-analysis consisted of reports providing data on the molecular detection of MβLs in *K. pneumoniae* isolates by PCR and/or WGS and describing the presence of at least two clinical isolates.

The quality assessment was carried out by two investigators (CSS and MJAS) using the Joanna Briggs Institute (JBI) critical appraisal checklist for analytical cross-sectional studies (0–8). Only when the conditional answer was “Yes” were the scores for completing the checklist questions considered [15,16].

### 2.4. Statistical Analysis

The Comprehensive Meta-Analyses—CMA program, version 3.3 (Biostat, Englewood, NJ, USA) and OpenMeta Analyst v.11 (Center for Evidence-Based Medicine, Brown University, Providence, RI, USA) were used to carry out the statistical analysis of the meta-analysis [17]. The random effects model (Der-Simonian Laird—DL method) estimated the combined frequency of *K. pneumoniae* with 95% confidence intervals (95% CI). A subgroup analysis was performed to determine the frequency of MβL-KP isolates according to Brazilian regions. The Chi-squared and I^2^ tests were used to determine the statistical difference groups (*p* < 0.05 was considered statistically significant). Egger and Begg’s rank correlation test and a funnel plot were used to examine the potential for publication bias (*p* < 0.05 was considered statistically significant). Sensitivity analysis, meta-regression, and subgroup analysis based on the study site were used to assess possible causes of variability, where applicable [18].

## 3. Results

### 3.1. Literature Search

After the initial screening and application of eligibility and inclusion criteria to the 496 records, 427 studies were excluded because their titles were not related to the topic, and 42 were excluded as duplicates, editor’s letters, brief communications, or incomplete studies. Following full-text assessment, 12 additional studies were excluded. The sampling process is depicted in Figure 1 below.

### 3.2. Characterization of the Included Studies

A total of 15 studies met our inclusion criteria and were retained for quantitative synthesis. All were published in English (15/15; 100%) and originated from Brazilian teaching and research institutions (15/15; 100%). In terms of database indexing, every study was retrievable via PubMed (15/15; 100%), with smaller subsets also found in SciELO (2/15; 13.3%) and ScienceDirect (1/15; 6.7%). All 15 investigations employed a retrospective, cross-sectional design (15/15; 100%).

Geographically, three (3) studies were conducted in the southeast region, four (4) in the northeast, two (2) in the north, two (2) in the mid-west, one (1) in the south, and (3) three sampled isolates from multiple macro-regions [19,20,21,22,23,24,25,26,27,28,29,30,31,32,33]. Study sample sizes ranged widely from 16 to 2.109 isolates. All studies investigated clinical isolates, primarily from tertiary-care hospitals or multi-center microbiology laboratories, and reported a variety of specimen types, including urine, blood, respiratory secretions, catheter tips, surgical wounds, and cerebrospinal fluid. Excluding studies not providing data on the isolate source, isolation from rectal/perianal swab were consistently reported as a sampling method.

Molecular detection of carbapenemase genes was uniformly performed by PCR in all 15 studies; two studies additionally applied whole-genome sequencing (WGS). Quality assessment via the Joanna Briggs Institute checklist yielded uniformly high scores in all the studies, indicating low risk of bias and high methodological quality across the included investigations (Table 1).

### 3.3. Prevalence of MβL-Harboring Isolates, Results, and Publication Bias of Proportion Meta-Analysis

Across the 15 included studies, a total of 3.533 *K. pneumoniae* isolates were screened for MβLs genes, of which 402 (11.4%) tested positive. The *bla*_NDM_ gene was overwhelmingly dominant, appearing in 14 of 15 studies with prevalence estimates spanning from 0.09% to 100%. By comparison, *bla*_VIM_ was detected in only two studies, and *bla*_IMP_ was exceptionally rare, found in just one isolate in a study by dos Santos et al. Under a random-effects model, the pooled MβLs prevalence was 44.6% (95% CI 24.2–65.1%; Q = 38 938.2, df = 14, *p* < 0.001; I^2^ = 99.96%), reflecting extreme between-study heterogeneity (Figure 2). Begg’s rank correlation test showed no evidence of publication bias (*p* = 0.96), corroborated by a symmetrical funnel plot (Figure 3).

Subgroup analyses were performed by Brazilian macro-region, revealing marked geographic heterogeneity in the prevalence of *K. pneumoniae* harboring MβLs genes. In the Southeast (three studies), the pooled prevalence was 59.8% (95% CI 3.6–115.9%); in the Midwest (two studies), 54.1% (95% CI −32.0–140.2%); in the Northeast (four studies), 37.8% (95% CI 15.8–59.7%); the South (two studies) showed a lower prevalence of 11.2% (95% CI 4.3–18.2%); and the North (two studies), 49.0% (95% CI −46.8–144.8%).

### 3.4. bla_NDM_ and bla_KPC_-Harboring Isolates

In the same set of studies that reported *bla*_NDM_, we also extracted *bla*_KPC_ data to allow direct comparison of these two carbapenemase genes. The pooled prevalence of *bla*_NDM_-positive *K. pneumoniae* was 41.7% (95% CI 20.6–62.8%), whereas that of *bla*_KPC_-positive isolates was 32.5% (95% CI 26.2–38.9%) (Figure 4 and Figure 5).

### 3.5. Meta-Regression Demonstrating the Impact of Study Size on MβL Prevalence

We examined the reason that residual heterogeneity remained extremely high (I^2^ = 99.96%) and, in this context, the factor of sample size of studies was used to try to explain this high heterogeneity in MβL prevalence by regressing logit-transformed rates on the number of isolates per study. There was a strong significant negative association with sample size (β = −0.0035 per isolate; *p* < 0.001), meaning smaller studies tended to report higher MβL rates (Figure 6).

## 4. Discussion

Epidemiologically, KPC and NDM have garnered significant attention due to their rapid global dissemination following their initial descriptions [34,35]. Data from the Pan-American surveillance network, covering 2015 to 2020, reinforce these concerns by showing a notable increase of 24.3% in *bla*_NDM_ carriage every six months [36]. In line with these findings, our meta-analysis estimated an overall pooled prevalence of 44.6% (95% CI 24.2–65.1%) for MβL-producing *K. pneumoniae* isolates in Brazil. Within this group, *bla*_NDM_-positive isolates accounted for 41.7% (95% CI 20.6–62.8%), and *bla*_KPC_-positive isolates represented 32.5% (95% CI 26.2–38.9%). These estimates are comparable to Brazilian national antimicrobial resistance surveillance data from 41,224 *K. pneumoniae* isolates (2015–2022), reporting a rising *bla*_NDM_ detection rate from 4.1% to 39.4%, alongside a decline in *bla*_KPC_ prevalence from 74.5% to 55.1%. Collectively, these findings highlight a clear epidemiological shift characterized by the declining dominance of KPC enzymes and the rapid emergence of NDM. Conversely, *bla*_VIM_ (1.3%) and *bla*_IMP_ (2.0%) remained uncommon and stable during the same period, reflecting distinct, lower-intensity transmission dynamics compared to *bla*_KPC_ and *bla*_NDM_ [11].

Our meta-analysis identified an overall MβL prevalence higher than figures reported in recent systematic reviews and meta-analyses from other global regions. For instance, Nasiri et al. [37] found a notably lower prevalence (24%) of CR-Kp in Iran, driven predominantly by *bla*_OXA-48_ rather than by MβLs. Similarly, another recent meta-analysis from Iran by Sadeghi et al. [38] reported even lower detection rates for MβLs, including *bla*_NDM_ (7.1%), *bla*_VIM_ (1.9%), and *bla*_IMP_ (0.9%), highlighting marked regional variability within the Middle East itself. Sisay et al. [39] documented a pooled prevalence of 15.1% for *bla*_NDM-1_ among clinical isolates in Africa, while Somda et al. [40] reported an even lower prevalence (10.6%) in West Africa.

In contrast, our Brazilian estimates align more closely with Asian data. For example, Dadashi et al. [41] found a pooled *bla*_NDM-1_ prevalence of 32.5% among Asian *K. pneumoniae* isolates. Likewise, Ain et al. [42] described a high prevalence (34.0%) of MβL among clinical isolates in Pakistan, with *bla*_NDM_ being the dominant variant, suggesting epidemiological similarities between Brazil and parts of South Asia. At a global level, Alvisi et al. [43] recently described the epidemiology of carbapenemase-producing Enterobacterales, reporting an overwhelming global predominance of *bla*_NDM_ (88.4% of all MβLs), followed by *bla*_VIM_ (11.1%) and *bla*_IMP_ (0.5%). While this global review reinforces our findings regarding NDM dominance, Brazil’s notably higher prevalence likely reflects distinct regional factors such as antibiotic prescribing patterns, healthcare infrastructure variability, and the effectiveness of infection-control measures. Such stark geographical contrasts highlight the importance of region-specific molecular surveillance and targeted intervention strategies.

Co-harboring of *bla*_KPC_ and *bla*_NDM_ was reported in 9 of our 15 studies (60%), spanning all five Brazilian macro-regions. In the southeast, Aires et al. [20] and Flores et al. [21] described isolates co-harboring *bla*_KPC-2_ and *bla*_NDM-1_ (often alongside ESBLs TEM/SHV/CTX-M). The northeast studies by Firmo et al. [22], Vivas et al. [23], and Oliveira et al. [24] similarly documented dual KPC + NDM producers. In the north, Rodrigues et al. [27] recovered both *bla*_KPC-2_+ *bla*_NDM-7_ and *bla*_NDM-1_ + *bla*_KPC-2_ strains, and Raro et al. [30] in the South reported KPC + NDM co-production alongside CTX-M. The multi-regional survey by Silveira et al. [31] even identified *K. quasipneumoniae* carrying both enzymes.

Nationally, ANVISA confirms that dual KPC + NDM producers first emerged in 2015 and increased 382.5% between 2018 and 2021. In that period, *Klebsiella* spp. accounted for 59.5% of all reported KPC + NDM co-producers, underscoring that this phenomenon is not confined to isolated outbreaks but is spreading across Brazilian healthcare settings. Crucially, this period overlaps with the COVID-19 pandemic, during which hospitals faced unprecedented strains [44]. These pandemic-related pressures likely fueled the accelerated dissemination of dual-enzyme-producing *K. pneumoniae*, compounding the threat posed by these highly resistant pathogens. The convergence of two potent carbapenemases on single plasmids leaves clinicians with virtually no β-lactam options and highlights the urgent need for multiplex molecular diagnostics, enhanced infection control, and stewardship interventions to halt further dissemination.

Although *bla*_NDM-1_ was the overwhelmingly predominant MβL variant, detected in 13 of the 14 NDM-positive studies, Rodrigues et al. [27] and Camargo et al. [32] reported the *bla*_NDM-7_ allele among *K. pneumoniae* isolates from the north and southeast regions, suggesting that, although NDM-1 remains the overwhelmingly predominant variant nationally, local pockets of NDM-7 expansion can and do arise, driven by high-risk clones and specific hospital networks. Distinguishing NDM-7 from NDM-1 is critical because NDM-7′s characteristic Asp130Asn and Met154Leu substitutions enhance carbapenem hydrolysis and often raise minimum inhibitory concentrations, signaling potentially higher resistance [45]. Moreover, different NDM variants can co-occur with unique arrays of co-resistance genes on diverse plasmid backbones, influencing both the breadth of antimicrobial resistance and the choice of effective therapies. Early and precise allele differentiation therefore underpins targeted infection-control and treatment measures.

High-risk clonal lineages, particularly ST11, have played a crucial role in the dissemination of MβLs genes, among clinical isolates of *K. pneumoniae* in Brazil [1,32]. These clones are associated with increased virulence, epidemic potential, and extensive multidrug resistance profiles, facilitating rapid and widespread dissemination in healthcare settings. Recent genomic studies conducted across Brazil consistently identify ST11 as a prominent clone harboring *bla*_NDM_ genes, often carried on highly transferable plasmids like IncF and IncX3, enhancing their dissemination potential through horizontal gene transfer. From south to north of Brazil, ST11 alongside other international high-risk clones (e.g., ST15 and ST340) have been frequently detected in relation to hospital outbreaks, further reinforcing their central role in propagating NDM carbapenemases regionally [27,46]. Collectively, these findings highlight that targeted surveillance and interventions aimed at controlling these high-risk clones are critical to mitigating the spread of MβL-producing *K. pneumoniae* in Brazil.

Our analysis revealed regional disparities in the prevalence of MβL-producing *K. pneumoniae* across Brazil, disparities that stem more from differences in study coverage and environmental factors than from true absence of resistance. Also, such studies may reflect differences in healthcare infrastructure, antimicrobial stewardship policies, laboratory capacity, and outbreak history. Outbreak investigations, launched in response to sudden clusters, tend to report the highest rates, as seen in Pereira et al. [19], who found *bla*_VIM_ in 61.9% of ICU isolates during a single-hospital event. By contrast, multicenter molecular-surveillance efforts that screen all clinical and screening swabs over time produce more moderate figures: Silveira et al. [31] detected *bla*_NDM_ in 21.5% of 205 isolates collected across several hospitals. Finally, passive clinical-isolate series, which analyze only specimens submitted for routine testing, may vary dramatically. Dos Santos et al. [26] recovered MβL genes in just 0.09% of 2 109 unselected isolates, whereas Rodrigues et al. [27] identified *bla*_NDM_ in all 23 consecutive clinical samples.

The inverse association between study size and reported MβL prevalence indicates a small-study effect, where smaller cohorts yield more extreme estimates. This phenomenon, common in microbiological prevalence studies, may stem from publication bias toward striking findings or from outbreaks investigated in single centers. Although Begg’s test did not detect significant bias, future research should prioritize multicenter designs with adequate sample sizes to mitigate this effect and provide more stable prevalence estimates.

Our findings should be viewed considering key limitations. First, variation in study sample size drove much of the between-study heterogeneity: Small, single-ward investigations frequently produced extreme prevalence estimates, whereas large multicenter series tended to dampen those peaks. Second, even after adjusting for sample size, residual heterogeneity remained extremely high, suggesting additional unmeasured moderators—such as study period, clinical setting (ICU versus surveillance samples), diagnostic technique (PCR versus WGS), and source of isolates—also influence the results. Third, by excluding letters and brief communications, we may have underestimated the true upper bounds of resistance. Fourth, our reliance on a narrowly defined set of search terms raises the possibility that relevant publications were overlooked. Despite these constraints, this analysis represents the first systematic synthesis of class B carbapenemase prevalence in *K. pneumoniae* across all Brazilian macro-regions. We adhered to PRISMA guidelines, performed dual independent data extraction, applied the Joanna Briggs Institute critical appraisal tool, and used multiple analytic strategies—including subgroup analyses and meta-regression—to probe heterogeneity. Collectively, these measures enhance reliability and transparency of our findings.

The striking regional variation in carbapenemase frequencies highlights the necessity of geographically customized infection-control strategies and antimicrobial-stewardship programs. Overall, the majority of MβL-producing *K. pneumoniae* isolates exhibited MDR or XDR phenotypes, with high resistance rates to β-lactams, aminoglycosides, and fluoroquinolones. From a clinical standpoint, these findings also highlight the urgent need for empiric therapy protocols in Brazilian hospitals to be informed by local carbapenemase epidemiology. In high-prevalence regions for MβL-producing *K. pneumoniae*, empiric regimens for severe infections may require inclusion of agents with reliable activity against these pathogens, such as novel β-lactam/β-lactamase inhibitor combinations or polymyxins, depending on susceptibility patterns. The frequent co-harboring of *bla*_NDM_ with *bla*_KPC_ further complicates treatment choices, as such strains often exhibit resistance to nearly all β-lactams [23]. Expanding laboratory infrastructure to include rapid molecular assays will be essential for timely identification of these resistance mechanisms and its variants. Prospective, multicenter surveillance using harmonized study protocols is urgently needed to confirm our prevalence estimates, uncover additional sources of heterogeneity, and guide public-health interventions.

Finally, although our review focuses exclusively on clinical *K. pneumoniae* isolates, recent environmental surveillance in Brazil has detected carbapenemase-producing *K. pneumoniae* in nonclinical settings. For instance, XDR *K. pneumoniae* strains harboring *bla*_KPC_, *bla*_NDM_, and *bla*_OXA-370_ have been isolated from hospital and municipal wastewater in Brazil [47]. These findings point out the environmental reservoirs of MβL-producing *Klebsiella*, reinforcing the need for integrated One Health surveillance strategies.

## 5. Conclusions

In this first nationwide synthesis of class B carbapenemase prevalence in Brazilian *K. pneumoniae* clinical isolates based on meta-analysis, we demonstrate that MβL producers, driven overwhelmingly by NDM variants, are both common and unevenly distributed across the country’s macro-regions. Co-harboring of *bla*_KPC_ and *bla*_NDM_, coupled with the emergence of a risk *bla*_NDM-7_, underscores the dynamic and convergent evolution of carbapenem resistance in Brazilian hospitals.

The very high residual heterogeneity and small-study effects reveal that prevalence estimates are heavily influenced by local factors, study size, clinical setting, diagnostic methods, and sampling strategies, highlighting the need for large, multicenter surveys using standardized molecular protocols. Exclusion of short communications and reliance on indexed databases may have led us to underestimate the true extremes of MβL dissemination. Although this meta-analysis does not provide prescriptive treatment recommendations, its epidemiologic signal supports hospital policies (diagnostics, screening, stewardship) that ultimately shape diagnostic and therapeutic pathways. Finally, sustained investment in high-quality molecular surveillance, coupled with strong antimicrobial stewardship, is essential to forestall further spread of these last-line resistance determinants.

## Figures and Tables

**Figure 1 antibiotics-14-00951-f001:**
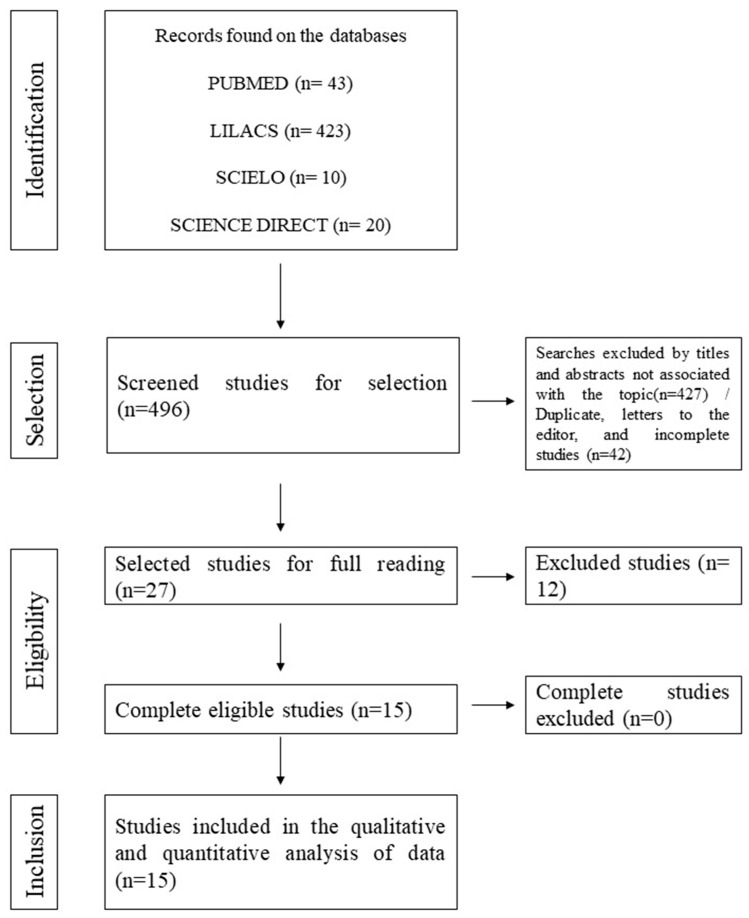
Flowchart of the data collection and screening steps for MβLs studies.

**Figure 2 antibiotics-14-00951-f002:**
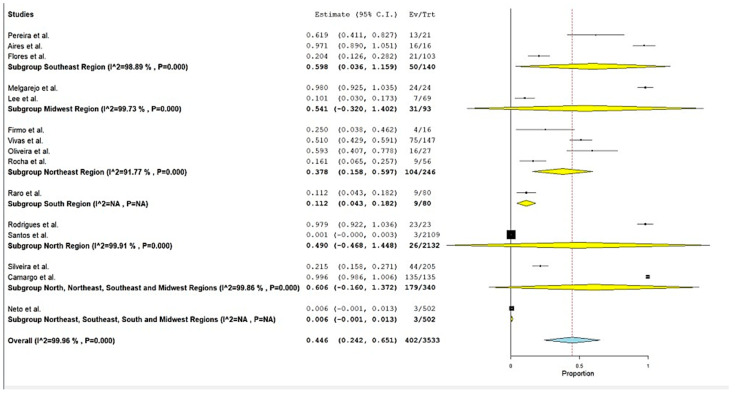
Forest plot of MβL prevalence among *K. pneumoniae* clinical isolates in Brazil, stratified by Brazilian macro-regions [19,20,21,22,23,24,25,26,27,28,29,30,31,32,33]. Legend: Diamonds represent pooled prevalence estimates with 95% confidence intervals (CI). Squares indicate point estimates from individual studies, with the size proportional to study weight in the meta-analysis. Horizontal lines show the 95% CI for each study.

**Figure 3 antibiotics-14-00951-f003:**
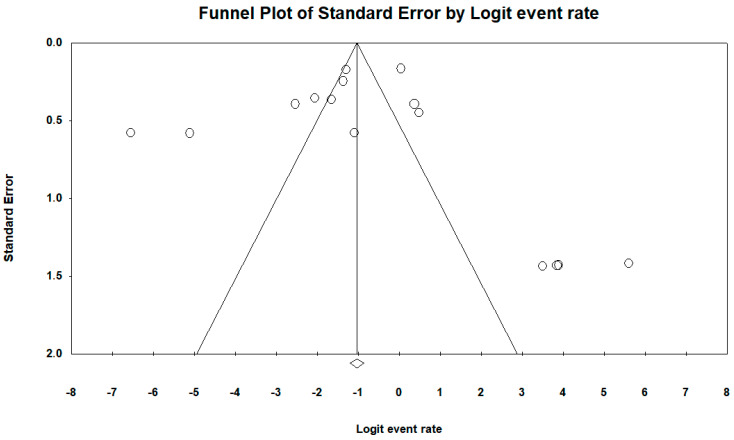
Funnel plot assessing publication bias among studies included in the meta-analysis of MβLs prevalence in *K. pneumoniae* isolates. Legend: The included published studies are shown as circles, which should be evenly spaced around the overall effect to form an inverted funnel. The most precise studies appear in the narrowest part of the funnel and lie closest to the true value. The standard error—plotted on the Y-axis as a measure of dispersion—is influenced by each study’s sample size; the larger the standard error, the more imprecise the study. The central line of the plot, marked on the X-axis by a diamond, represents the effect measure result examined in the meta-analysis. The lines outlining the funnel correspond to the 95% confidence intervals.

**Figure 4 antibiotics-14-00951-f004:**
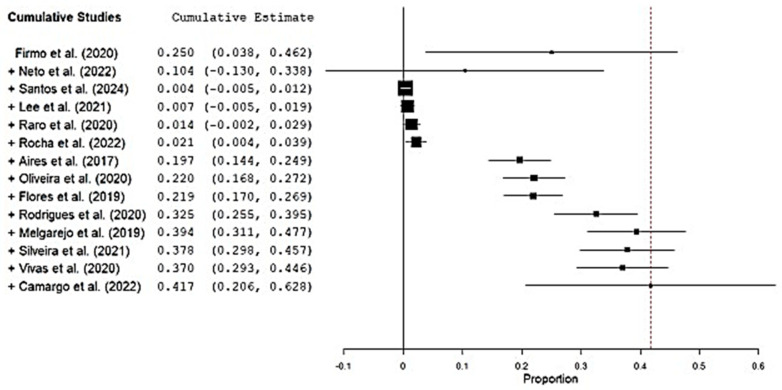
Forest plot of the proportion of *bla*_NDM_-positive *K. pneumoniae* isolates reported in individual studies [20,21,22,23,24,25,26,27,28,29,30,31,32,33]. Legend: The pooled prevalence estimate is represented by the diamond at the bottom of the plot, with its width corresponding to the 95% confidence interval (CI). Squares indicate point estimates for each study, with the size proportional to the study’s weight in the meta-analysis. Horizontal lines show the 95% CI for each study.

**Figure 5 antibiotics-14-00951-f005:**
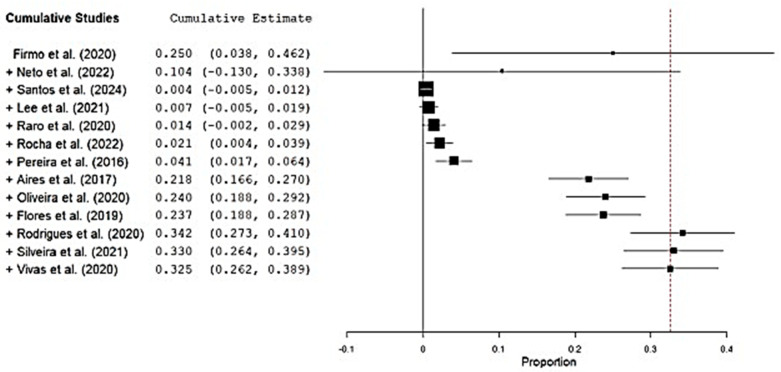
Forest plot of the proportion of *bla*_KPC_-positive *K. pneumoniae* isolates reported in individual studies [19,20,21,22,23,24,25,26,27,29,30,31,33]. Legend: The pooled prevalence estimate is represented by the diamond at the bottom of the plot, with its width corresponding to the 95% confidence interval (CI). Squares indicate point estimates for each study, with the size proportional to the study’s weight in the meta-analysis. Horizontal lines show the 95% CI for each study.

**Figure 6 antibiotics-14-00951-f006:**
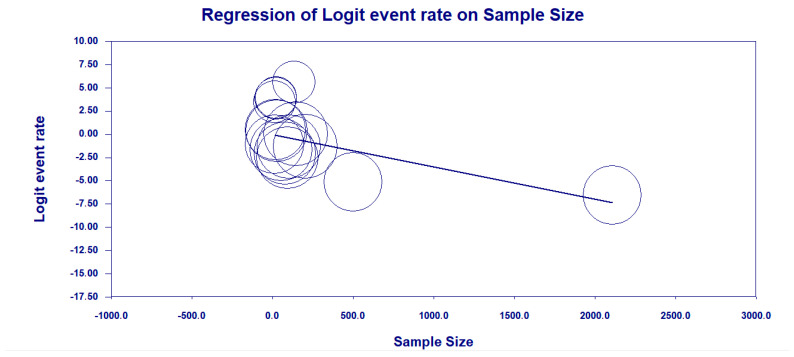
Scatterplot of the meta-regression of the logit-transformed MβL-KP event rate against study sample size (n = number of isolates per study). Legend: Bubble areas are inversely proportional to each study’s variance (larger bubbles = greater weight). The solid line represents the fitted regression (β = −0.0035 per isolate; *p* < 0.001), demonstrating that smaller studies tend to report higher logit event rates of MβL-KP.

**Table 1 antibiotics-14-00951-t001:** Summary of included studies: Period, Location, Isolate Sources, and MβL Prevalence.

N°	Authors/Year	Type of Study/Sample Period	Location/ Region	Setting Profile	N° of Kp Isolates/N° Isolates Tested	Isolate Source	Results (MβLs)	Results (*bla*_KPC_)	Carbapenemases Detection	JBI Score
1	Pereira et al. [19] *	Retrospective cross-sectional/January–December 2012	Juiz de Fora, Minas Gerais State/southeast region	Clinical–one tertiary-care hospital	1076/21	Urine, blood, tracheal secretion, bronchoalveolar lavage, and catheter tip	*bla*_VIM_ prevalence (N = 13/21; 61.9%)	*bla*_KPC_ positive (20/21)	PCR	(8/8)
2	Aires et al. [20] *	Retrospective cross-sectional/July 2013–November 2014	Rio de Janeiro, Niterói, Campos dos Goytacazes Rio de Janeiro State/southeast region	Clinical/Eight healthcare institutions	16/16	Rectal swabs, urine, blood, catheter tip, and CSF	*bla*_NDM_ prevalence (N = 16/16; 100%)	*bla*_KPC_ positive (2/16)	PCR	(8/8)
3	Flores et al. [21] *	Retrospective cross-sectional/March 2016–March 2017	Rio de Janeiro State/Southeast region	Clinical/Tertiary-care hospital ICU	103/103	Rectal swabs	*bla*_NDM_ prevalence (N = 11/103; 10.7%) *bla*_VIM_ prevalence (N = 10/103; 9.7%)	*bla*_KPC_ positive (8/103)	PCR	(8/8)
4	Firmo et al. [22] *,#	Retrospective cross-sectional/2016–2018	Recife, Pernambuco State/Northeast region	Clinical/Three public and private hospitals	16/16	Urine, surgical wound, tracheal aspirate, rectal swab blood, CSF, bone	*bla*_NDM_ prevalence (n = 4/16; 25.0%)	*bla*_KPC_ positive (15/16)	PCR	(8/8)
5	Vivas et al. [23] *^,@^	Retrospective cross-sectional	Aracaju, Sergipe state/northeast region	Clinical/Hospital microbiology laboratory	147/147	Not described	*bla*_NDM_ prevalence (n = 75/147; 51.02%)	*bla*_KPC_ positive (9/147)	PCR	(8/8)
6	de Oliveira et al. [24] *^,@^	Retrospective cross-sectional/2017–2018	Recife, Pernambuco state/northeast region	Clinical/Public hospital	27/27	Different sites on infection and colonization	*bla*_NDM_ prevalence (n = 16/27)	*bla*_KPC_ positive (24/27)	PCR	(8/8)
7	Rocha et al. [25] *	Retrospective cross-sectional/2016–2018	Bahia state/northeast region	Clinical/Tertiary care hospital	56/56	Rectal swab, blood culture, catheter tip, intraoperative secretion, tracheal secretion, and urine culture	*bla*_NDM_ prevalence (n = 9/56; 16.07%)	*bla*_KPC_ positive (53/56)	PCR	(8/8)
8	dos Santos et al. [26] *	Retrospective cross-sectional/	Rondônia state/north region	Clinical/Central public health laboratory (LACEN/RO)	2.109/2.109	Not described	*bla*_NDM_ prevalence (n = 2/2109; 0.09%) *bla*_IMP_ prevalence (n = 1/2109; 0.05%)	*bla*_KPC_ positive (272/2109)	PCR	(8/8)
9	Rodrigues et al. [27] *	Retrospective cross-sectional/2018–2021	Belém, Pará state/north region	Clinical/Nine different healthcare institutions in the region with clinical wards (non-ICU settings) and ICU isolates	23/23	Urine, blood, tracheal secretion, rectal swab, bronchoalveolar lavage, abdominal abscess secretion, wound secretion, nasopharyngeal secretion, soft-tissue secretion, and peritoneal fluid	*bla*_NDM_ prevalence (n = 23/23; 100%),	*bla*_KPC_ positive (4/23)	PCR	(8/8)
10	Melgarejo et al. [28] *	Retrospective cross-sectional/2013–2014	Brasília, Distrito Federal, mid-west region	Clinical/Ten distinct hospitals	24/24	Urine, rectal swab, blood, and tracheal aspirates	*bla*_NDM_ prevalence (N = 24/24; 100%)	Not reported	PCR	(8/8)
11	Lee et al. [29] *	Retrospective cross-sectional/2010–2014	Brasília, Distrito Federal, mid-west region	Clinical/Twelve hospitals	95/69	Not described	*bla*_NDM_ prevalence (n = 7; 10.14%)	*bla*_KPC_ positive (61/69)	WGS	(8/8)
12	Raro et al. [30] *	Retrospective cross-sectional/2017–2018	Porto Alegre, Rio Grande do Sul state, south region	Clinical/1.000 bed hospital complex	80/80	Rectal swabs	*bla*_NDM_ prevalence (n = 9/80; 11.25%)	*bla*_KPC_ positive (71/80)	WGS	(8/8)
13	Silveira et al. [31] *	Retrospective cross-sectional/2019–2020	Several locations/northeast, north, southeast, and mid-west regions	Clinical/Hospitals	205/205	Blood and catheter tip	*bla*_NDM_ prevalence (N = 44/205; 21.47%)	*bla*_KPC_ positive (142/205)	PCR	(8/8)
14	Camargo et al. [32] *	Retrospective cross-sectional/2013–2021	Several locations/northeast, north, southeast, and mid-west regions	Clinical/Hospitals	135/135	Surveillance swab (rectal or perianal), urine, blood or catheter tip, upper respiratory tract secretion, CSF, other clinical samples	*bla*_NDM_ prevalence (N = 135/135; 100.0%)	Not reported	PCR and WGS	(8/8)
15	Conceição-Neto et al. [33] *	Retrospective cross-sectional/	Several locations/northeast, mid-west, southeast, and south regions	Clinical/Hospitals	502/502	Blood, urine, tracheal aspirate, rectal swab, catheter tip, sputum, tissue fragment, and wound	*bla*_NDM_ prevalence (n = 3/502; 0.59%)	*bla*_KPC_ positive (122/502)	PCR	(8/8)

Legend: * Retrieved from PubMed database based on search strategy; # retrieved from Science Direct database based on search strategy; @ retrieved from ScieLO database.

## Data Availability

The original contributions presented in the study are included in the article. Further inquiries can be directed to the corresponding authors. PROSPERO registration can be accessed at: https://www.crd.york.ac.uk/PROSPERO/view/CRD420251110931 (accessed on 23 July 2025).

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
