# Peer review of "Nationwide Burden of Metallo-β-Lactamase Genes in Brazilian Clinical *Klebsiella pneumoniae* Isolates: A Systematic Review and Meta-Analysis"

_antibiotics, 2025, doi:10.3390/antibiotics14090951_

Round 1
Reviewer 1 Report
Comments and Suggestions for Authors
- In discussion; Please add data about the occurrence of ESBL genes in the reported strains
- In discussion; Please add data about antibiotic susceptibility of the reported strains (MDR, XDR)
- In discussion; Please add data about detected plasmids, integrons occurrence (as well as gene cassette arrays)
- In discussion; can you provide data about the prevalence of MBL in other Enterobacteriaceae genus?
- In discussion, please provide in few sentence data about MBL in livestock or water sources (I prefer to put this article in the context of one health), this make the article more interesting for reader out side Brazil
Author Response
Author's Reply to the Review Report (Reviewer 1)
In discussion; Please add data about the occurrence of ESBL genes in the reported strains
Reply: We thank the reviewer for this suggestion. While not all included studies reported ESBL data, the occurrence of ESBL genes in MβL-producing K. pneumoniae is already addressed in our Discussion. For instance, co-harboring of KPC and NDM was reported in nine of our 15 studies (60%), with several also detecting ESBL genes. These examples provide representative evidence of ESBL co-occurrence. We decided not to expand this section further in order to avoid a lengthy discussion that might detract from the primary focus of the manuscript, which is the national epidemiology of MβL genes.
In discussion; Please add data about antibiotic susceptibility of the reported strains (MDR, XDR)
Reply: Included as requested.
Lines 351 – 353: Overall, the majority of MβL-producing K. pneumoniae isolates exhibited MDR or XDR phenotypes, with high resistance rates to β-lactams, aminoglycosides, and fluoroquinolones.
In discussion; Please add data about detected plasmids, integrons occurrence (as well as gene cassette arrays)
Reply: We thank the reviewer for this suggestion. Not all included studies reported data on plasmid types, integron occurrence, or gene cassette arrays. However, where available, these elements are already discussed in the manuscript — for example, the detection of IncF and IncX3 plasmid replicon types, class 1 integrons, and associated resistance gene cassettes. Given that these mentions already capture the genetic platforms identified in the reviewed studies, and to avoid a lengthy discussion that might detract from the manuscript’s main focus, we have opted not to expand this section further.
In discussion; can you provide data about the prevalence of MBL in other Enterobacteriaceae genus?
Reply: We thank the reviewer for this suggestion. The prevalence of MβL genes in other Enterobacteriaceae genera was not the focus of our study, which was designed specifically to assess K. pneumoniae. Although some included studies occasionally reported MβL genes in species such as Enterobacter cloacae, Escherichia coli, and Serratia marcescens, these findings were incidental and not systematically captured in our analysis. Given this limited and heterogeneous reporting, and to maintain the manuscript’s defined scope, we have opted not to expand this section further.
In discussion, please provide in few sentence data about MBL in livestock or water sources (I prefer to put this article in the context of one health), this make the article more interesting for reader out side Brazil
Reply: Included as requested.
Lines 365 – 371: “Finally, although our review focuses exclusively on clinical K. pneumoniae isolates, recent environmental surveillance in Brazil has detected carbapenemase producing K. pneumoniae in nonclinical settings. For instance, XDR K. pneumoniae strains harboring blaKPC, blaNDM, and blaOXA 370 have been isolated from hospital and municipal wastewater in Brazil [47]. These findings point out the environmental reservoirs of MβL-producing Klebsiella, reinforcing the need for integrated One Health surveillance strategies.”
Reviewer 2 Report
Comments and Suggestions for Authors
This manuscript addresses an important and timely topic in antimicrobial resistance: the prevalence and distribution of MβL genes in K. pneumoniae from Brazil. The focus on a national-level synthesis through systematic review and meta-analysis fills a significant knowledge gap, given the sporadic nature of available data. The study is well-motivated, relevant for public health, and clearly within the scope of microbiology and infectious diseases.
The methods are generally well-described, and the statistical approach is appropriate for the heterogeneity expected in epidemiological studies. The conclusions are consistent with the results and emphasize the need for enhanced surveillance and laboratory capacity.
Author Response
Author's Reply to the Review Report (Reviewer 2)
This manuscript addresses an important and timely topic in antimicrobial resistance: the prevalence and distribution of MβL genes in K. pneumoniae from Brazil. The focus on a national-level synthesis through systematic review and meta-analysis fills a significant knowledge gap, given the sporadic nature of available data. The study is well-motivated, relevant for public health, and clearly within the scope of microbiology and infectious diseases.
The methods are generally well-described, and the statistical approach is appropriate for the heterogeneity expected in epidemiological studies. The conclusions are consistent with the results and emphasize the need for enhanced surveillance and laboratory capacity.
Author’s Response:
We sincerely thank the reviewer for their thoughtful and encouraging assessment of our work. We appreciate the recognition of the study’s relevance, methodological soundness, and alignment between results and conclusions.
Reviewer 3 Report
Comments and Suggestions for Authors
Dear Authors,
The number of studies were 15 for the actual analysis. the limitations of studies are discussed poorly and in my opinion the this study is limited because of its actual clinical significance which can not be applied for improvement of patient's disease diagnosis or treatment.
Author Response
Author's Reply to the Review Report (Reviewer 3)
Dear Authors,
The number of studies were 15 for the actual analysis. the limitations of studies are discussed poorly and in my opinion the this study is limited because of its actual clinical significance which can not be applied for improvement of patient's disease diagnosis or treatment.
Author’s Response:
We thank the reviewer for their comments. While we agree that clinical applicability is an important consideration, we respectfully note that the primary objective of this work was to synthesize epidemiological data on the prevalence and geographic distribution of MβL genes in Brazilian K. pneumoniae isolates through a systematic review and meta-analysis. This type of national-level synthesis is intended to inform public health strategies, laboratory surveillance priorities, and infection control policies, rather than to serve as a direct interventional tool for individual patient management.
Regarding the number of studies included (n = 15), this reflects the totality of eligible Brazilian publications meeting our strict inclusion criteria over the 2006–2024 period, as determined by our protocol. This number is comparable to other national or regional meta-analyses in the antimicrobial resistance field, given the specificity of the resistance mechanism and organism under study.
We would also like to highlight that our discussion already addresses limitations, (including study size variability, heterogeneity, and lack of uniform diagnostic approaches) which inherently constrain direct clinical application. Nonetheless, the findings contribute to a foundational knowledge base that can support the development of targeted interventions, which, in turn, can indirectly improve diagnosis and treatment through enhanced surveillance and evidence-based empiric therapy guidelines.
Reviewer 4 Report
Comments and Suggestions for Authors
Dear Authors,
The presented study is a thorough and well-written review. I read it with interest and I find it important and publishable.
There are a few remarks that I recommend addressing:
- Line 34: Consider rephrasing “urgent expansion of molecular surveillance” to specify what type of surveillance is needed (e.g., multicenter, longitudinal, allele-specific).
- Line 90: The POT strategy is mentioned but not widely recognized. A brief explanation or citation would help readers unfamiliar with this framework.
- Table 1 is dense and difficult to navigate. Consider reformatting it for clarity—perhaps by grouping studies by region or highlighting key findings (e.g., highest prevalence, co-harboring of genes).
- Figures: Ensure all figures are high-resolution and include clear legends. The funnel plot and forest plots are informative but could be better labeled.
- Language: Minor grammatical and stylistic edits are needed throughout the manuscript to improve readability and flow.
- Gene names (e.g., bla, ndm, vim) should be italicized consistently throughout.
- The discussion would benefit from a more explicit connection to clinical practice. For example, how should these findings influence empiric therapy or infection control protocols in Brazilian hospitals?
In summary, the manuscript is of high quality and addresses a critical topic in antimicrobial resistance.
Please refer to the above remarks.
Comments on the Quality of English Language
The English language used needs a bit of refining. I chose the "The English could be improved to more clearly express the research", but the improvemens needed are minor, but still.
Author Response
Author's Reply to the Review Report (Reviewer 4)
Dear Authors,
The presented study is a thorough and well-written review. I read it with interest and I find it important and publishable.
There are a few remarks that I recommend addressing:
- Line 34: Consider rephrasing “urgent expansion of molecular surveillance” to specify what type of surveillance is needed (e.g., multicenter, longitudinal, allele-specific).
Reply: Corrected as requested.
- Line 90: The POT strategy is mentioned but not widely recognized. A brief explanation or citation would help readers unfamiliar with this framework.
Reply: We thank the reviewer for this observation. We have opted not to expand the description of the POT strategy within the Methods section to maintain conciseness and avoid redundancy. However, we have ensured that appropriate references are provided so that readers unfamiliar with the framework can consult the original sources for a detailed explanation.
- Table 1 is dense and difficult to navigate. Consider reformatting it for clarity—perhaps by grouping studies by region or highlighting key findings (e.g., highest prevalence, co-harboring of genes).
Reply: We appreciate the reviewer’s perspective regarding Table 1. However, we have chosen to retain the current structure, as it aligns with PRISMA guidelines for systematic reviews and meta-analyses, which recommend presenting key study characteristics in a standardized, comprehensive format to ensure transparency and reproducibility. Grouping or highlighting selected results within the table could inadvertently introduce emphasis bias.
- Figures: Ensure all figures are high-resolution and include clear legends. The funnel plot and forest plots are informative but could be better labeled.
Reply: Corrected as requested.
- Language: Minor grammatical and stylistic edits are needed throughout the manuscript to improve readability and flow.
Reply: We thank the reviewer for this observation. Minor grammatical and stylistic corrections have been made throughout the manuscript to enhance readability. Additionally, we note that further minor edits will be incorporated by the MDPI proofreading team as part of the journal’s standard editorial process prior to publication.
- Gene names (e.g., bla, ndm, vim) should be italicized consistently throughout.
Reply: Corrected as requested.
- The discussion would benefit from a more explicit connection to clinical practice. For example, how should these findings influence empiric therapy or infection control protocols in Brazilian hospitals?
Reply: We appreciate this important suggestion. In response, we have expanded the Discussion section to explicitly address the clinical implications of our findings for Brazilian hospitals.
Lines 340 – 346 : “From a clinical standpoint, these findings underscore the urgent need for empiric therapy protocols in Brazilian hospitals to be informed by local carbapenemase epidemiology. In high-prevalence regions for MβL-producing K. pneumoniae, empiric regimens for severe infections may require inclusion of agents with reliable activity against these pathogens, such as novel β-lactam/β-lactamase inhibitor combinations or polymyxins, depending on susceptibility patterns. The frequent co-harboring of blaNDM with blaKPC further complicates treatment choices, as such strains often exhibit resistance to nearly all β-lactams [23].”
Round 2
Reviewer 3 Report
Comments and Suggestions for Authors
Dear Authors,
The manuscript need rigorous amendments to make it more relevant for readers and journal's scope. In my opinion the previous comments are not addressed scientifically as per the proper scientific study.
Author Response
Thank you for the feedback.
We take relevance and scientific rigor seriously; however, the comment is broad and does not identify specific sections, methods, or results requiring amendment.
We have already presented the of the present study that explicitly detail small-study effects, extreme heterogeneity, varied sampling frames, and diagnostic non-uniformity (lines 337–352).
We also added a concise Clinical & Public-Health Implications paragraph that translates the surveillance signal into system-level actions without overstating patient-level impact. (Lines 355-366)
In response to you previous comment "this study is limited because of its actual clinical significance which can not be applied for improvement of patient's disease diagnosis or treatment" we explicitly state:
“Although this meta-analysis does not provide prescriptive treatment recommendations, its epidemiologic signal supports hospital policies (diagnostics, screening, stewardship) that ultimately shape diagnostic and therapeutic pathways” (lines 387–390)."
Finally, the manuscript’s aims and content clearly align with the journal’s scope on antimicrobial-resistance epidemiology and public-health utility, in line with comments by Editor of the Special Issue and other Reviewer.
If there are specific prior comments the reviewer believes remain unaddressed, we would be grateful for pointers to the exact items and manuscript locations so we can revise those sections precisely.